# Unraveling the Role of the Liriodendron Thioredoxin (TRX) Gene Family in an Abiotic Stress Response

**DOI:** 10.3390/plants13121674

**Published:** 2024-06-17

**Authors:** Lu Tong, Mengyuan Lin, Liming Zhu, Bojun Liao, Lu Lu, Ye Lu, Jinhui Chen, Jisen Shi, Zhaodong Hao

**Affiliations:** 1State Key Laboratory of Tree Genetics and Breeding, Co-Innovation Center for Sustainable Forestry in Southern China, Nanjing Forestry University, Nanjing 210037, China; tonglu202402@163.com (L.T.); 15990331823@163.com (M.L.); zhulm@njfu.edu.cn (L.Z.); 15779776739@163.com (B.L.); lulu2020@njfu.edu.cn (L.L.); luye@njfu.edu.cn (Y.L.); chenjh@njfu.edu.cn (J.C.); 2Key Laboratory of Forest Genetics & Biotechnology of Ministry of Education, Nanjing Forestry University, Nanjing 210037, China

**Keywords:** thioredoxin, *Liriodendron* hybrid, abiotic tolerance

## Abstract

Thioredoxin (TRX) is a small protein with REDOX activity that plays a crucial role in a plant’s growth, development, and stress resistance. The TRX family has been extensively studied in Arabidopsis, rice, and wheat, and so it is likely that its members have similar biological functions in *Liriodendron* that have not been reported in *Liriodendron*. In this study, we performed the genome-wide identification of the *TRX* gene family based on the *Liriodendron chinense* genome, leading to a total of 42 *LcTRX* gene members. A phylogenetic analysis categorized these 42 LcTRX proteins into 13 subfamilies. We further characterized their chromosome distributions, gene structures, conserved protein motifs, and cis-elements in the promoter regions. In addition, based on the publicly available transcriptome data for *Liriodendron* hybrid and following RT-qPCR experiments, we explored the expression patterns of *LhTRXs* to different abiotic stressors, i.e., drought, cold, and heat stress. Notably, we found that several *LhTRXs*, especially *LhTRX-h3*, were significantly upregulated in response to abiotic stress. In addition, the subcellular localization assay showed that LhTRX-h3 was mainly distributed in the cytoplasm. Subsequently, we obtained *LhTRX-h3* overexpression (OE) and knockout (KO) callus lines in *Liriodendron* hybrid. Compared to the wild type (WT) and *LhTRX-h3*-KO callus proliferation of *LhTRX-h3*-OE lines was significantly enhanced with reduced reactive oxygen species (ROS) accumulation under drought stress. Our findings that *LhTRX-h3* is sufficient to improve drought tolerance. and underscore the significance of the *TRX* gene family in environmental stress responses in *Liriodendron*.

## 1. Introduction

Under natural conditions, plants often suffer from environmental stressors during their growth and development, among which drought and water shortage are serious abiotic stressors that threaten plant production [1,2,3,4,5]. The process of a plant’s resistance to drought is very complex [3,6], and a large amount of evidence has shown that reactive oxygen species (ROS) production and removal are closely related to drought tolerance mechanisms [7,8,9]. When plants are under drought stress, reactive oxygen species are rapidly produced [10]. When a large amount of ROS accumulates in a plant, it will seriously affect the growth and development of that plant [10,11]. As a result, plants have evolved systems that regulate reactive oxygen species levels, such as the TRX system [12,13,14].

Thioredoxin (*TRX*) is a class of ubiquitous disulfide reductases with two active cysteine sites in the conserved motif WC[G/P]PC, namely, the CXXC motif [15], which is widely present in the tissue proteins of most organisms. The TRX class is divided into typical TRXs and atypical TRXs [16]. Typical TRXs are further divided into seven subfamilies (TRXh, TRXf, TRXm, TRXz, TRXx, TRXy, and TRXo) based on their amino acid sequence similarities and subcellular localizations, while atypical TRXs are divided into eight subfamilies (like1, like2, Nrx, Picot, clot, CDSP32, HCF164, and Lillium) [17].

The thioredoxin system, which consists of NADPH, thioredoxin reductase (TRXR), and thioredoxin, is a key antioxidant system that regulates the balance of protein dimercaptan/disulfide bonds through the activity of disulfide reductase to protect against oxidative stress [13,18]. The TRX system provides electrons to mercaptan-dependent peroxidase (a peroxido-reducing protein) to remove reactive oxygen species and nitrogen substances at a rapid reaction rate [12]. TRX proteins are versatile and play critical roles in translation, protein assembly, hormone synthesis, metabolism, plant development, seed germination, and oxidative stress responses to both biotic and abiotic stress [19,20].

It has been reported that *KcTRXf* transgenic plants help to scavenge ROS and regenerate GSH from oxidized glutathione more efficiently [21]. Malondialdehyde (MDA) is one of the commonly used indicators of oxidative stress, which reflects the degree of lipid peroxidation in plant membranes. Transgenic tobacco plants overexpressing *SlTRXh* showed lower levels of MDA and ROS accumulation under nitrate stress, suggesting that *SITRXh* may function under ROS homeostasis [22]. The silencing of the *ClTRXh2* gene made watermelon plants more sensitive to cold stress [23]. Thioredoxin plays an important role in a plant’s defense against oxidative stress.

Liriodendron is an ancient relict with only two species in its genus in nature, namely *L. chinense* and *L. tulipifera* [24]. *L. chinense* is mainly distributed in the south of the Yangtze River, and its growth is restricted by the unfavorable environmental conditions of high temperature, low temperature, and soil drought. TRX, as a key player in cellular REDOX homeostasis, is known to be involved in a plant’s abiotic stress response. There has been no report of *TRXs* in *L. chinense*; therefore, it is important to explore the role of TRX in a plant’s abiotic stress tolerance. In this study, we identified 42 LcTRXs in *L. chinense* for evolutionary analysis that considered basic physicochemical properties, a phylogenetic analysis, a gene structure analysis, a cis-acting element analysis, etc., and the expression patterns of the TRXs were quantitatively probed under cold stress, heat stress, and drought stress using RT-qPCR. In the process, an interesting gene, *TRX-h3*, was discovered, and its function under stress was preliminarily explored through its overexpression, which provided a basis for the further study of TRXs in *Liriodendron*.

## 2. Results

### 2.1. Identification and Protein Sequence Characterization of the LcTRXs

A total of 70 TRX-like protein sequences were obtained through Blastp and hmmersearch via CD-Search (https://www.ncbi.nlm.nih.gov/Structure/bwrpsb/bwrpsb.cgi, accessed on 5 February 2022). To further confirm the TRX domain and screen out TRX-like proteins with incomplete domains, we obtained 42 non-redundant sequences of TRXs for subsequent analysis (Table 1). Among the typical TRXs, there were 20 sequences categorized into six types: TRXh, TRXf, TRXm, TRXx, TRXy, and TRXz. Notably, the h-type was the most abundant, and they were subdivided into the I, II, and III subtypes, while the TRXx, TRXy, and TRXf type sequences were the least represented, with only one each. Atypical TRXs comprised 22 sequences across eight categories, including TRXlike, Lillium, CDSP32, Clot, HCF164, Nrx, Picot, and TDX, with Nrx being the most abundant, followed by Lillium. To elucidate the molecular characteristics of each LcTRX protein, we calculated the amino acid numbers, molecular weights (MVs), and isoelectric points (pls) of the proteins using tools on the ExPASy website. The results revealed that the lengths of the LcTRX proteins ranged from 94 to 910 amino acids, with molecular weights corresponding to the protein lengths and ranging from 10.539 kDa to 102.664 kDa. The theoretical isoelectric points of the LcTRXs ranged from 4.52 to 9.30. Interestingly, the typical TRX acidic proteins were half as prevalent as the basic proteins, whereas the atypical TRX basic proteins were only five in number, with the remainder being acidic proteins (Table 1).

### 2.2. Phylogenetic Analysis, Conserved Domains, and Gene Structure Analysis

In order to elucidate the molecular evolution and phylogenetic relationships of *L. chinense* TRX proteins, an unrooted phylogenetic tree was constructed containing 42 LcTRXs and their homologs in Arabidopsis, rice, and poplar. A multiple sequence alignment of TRX gene family members was performed using MAFFT software (v7.487) (https://mafft.cbrc.jp/alignment/software) (accessed on 15 November 2021) with default parameters. The phylogenetic tree was constructed using MAGE 7.0, employing the neighbor-joining method with a bootstrap value of 1000 to analyze the evolution of the TRX gene in *L. chinense*. The results revealed that the TRX homologs of *L. chinense* could be divided into 14 branches, with the h subgroups further divided into hI, hII, and hIII (Figure 1). By comparing the genomic DNA sequences, we were able to ascertain the exon and intron structures of LcTRXs. All exons were interrupted by introns, with the number of exons ranging from two to a maximum of seven. The majority of *LcNRX* genes exhibited structural similarities, with the exception of *LcNRX1*, which had a small number of exons, only one intron, and a very short length. Furthermore, the protein sequences of LcTRXs were analyzed using the online tool MEME (Figure 2). It was found that LcNRXs had the largest number and variety of motifs, but did not contain motif 4. The remaining Lc TRXs exhibited the presence of motifs 1 and 4, with the exception of LcTRXlike1 and LcTRXlike1.1, which exhibited the presence of motif 4 alone, and LcTRXlike2 and LcTRXlillium2, which exhibited the presence of motif 1 alone.

### 2.3. Chromosomal Location and Collinearity Analysis

By analyzing the chromosomal location and tandem duplication of genes, the results reveal that 37 gene sequences of *LcTRXs* are distributed on 12 chromosomes, with the largest number located on chromosome 1, and the fewest on chromosomes 10, 12, 13, and 18, each containing only one *LcTRX* gene (Figure 3A). Additionally, the remaining five genes are located on four contigs, respectively. Furthermore, we identified three pairs of tandem repeats. The Ka/Ks ratio is a measure of the evolutionary pressure of a species [25]. When Ka/Ks > 1, the gene pair is subject to positive selection; when Ka/Ks = 1, the gene pair is subject to neutral evolutionary selection; and when Ka/Ks < 1, the gene pair is subject to purifying selection. Here, we calculated the Ka/Ks of three pairs of tandem repeats, all of which were less than 1, indicating selection by purifying selection (Appendix A).

In order to further study the evolutionary status of LcTRXs in different species, the whole genomes of Arabidopsis, rice, and *L. chinense* were selected for collinearity analysis. Arabidopsis and rice belong to dicotyledonous and monocotyledonous plants, respectively. The results showed that Arabidopsis and *L. chinense* exhibited eight collinear relationships, while rice showed fifteen collinear relationships with *L. chinense*. This suggests that *L. chinense* is more closely related to dicotyledonous plants like Arabidopsis than to monocotyledonous plants like rice in the evolutionary process [26] (Figure 3B).

### 2.4. Cis-Acting Element Analysis

To further analyze the function of *TRXs*, the upstream 2.0 kb promoter sequence of *LcTRXs* was extracted, and the online website PlantsCARE was utilized for cis-acting element analysis. As a result, the LcTRX promoter region was found to be enriched with various types of cis-acting elements, including 373 hormone-responsive elements, 359 development-related elements, and 436 abiotic stress-responsive elements. Development-related elements (as-1) were found in 38 *LcTRXs*, while abiotic stress response elements included light-responsive elements (Box4, GT1-motif, I-box), drought-responsive elements (MBS), and wound-responsive elements (WUN-motif), as well as low-temperature response element (LTR) and oxidative stress response element (ARE). Nearly all *LcTRXs* contained light-responsive elements (including Box4, GT1-motif, I-box), with only four genes (*LcTRXh4*, *LcTRXy2*, *LcTRXx*, *LcNrx1.2*) lacking any light-responsive elements. Moreover, more than half of the genes contained drought response elements and corresponding elements for oxidative stress. Plant hormone-related response elements accounted for the largest proportion of all cis-acting elements, including abscisic acid response (ABRE), salicylic acid response (TCA-element), gibberellin response (P-box), auxin response (TGA-element, AuxRR-core), and methyl jasmonate reaction (CGGTA-motif). The results indicated that *LcTRXs* may play a role in plant growth and development, abiotic stress, and hormone response (Figure 4).

### 2.5. Expression Patterns under Abiotic Stress

In order to study the expression of *TRXs* under cold, heat, and drought stresses in *L.* hybrid, the laboratory’s previous RNA-seq data were used for analysis [26], and the results are shown in Figure 5. Different *LhTRXs* exhibit varying responses to different biological stresses. Under low-temperature stress, 17 *LhTRXs* reached their peak values at 3 d, include *LhClot*, *LhTRXy1*, *LhNrx1.2*, *LhTRXlike2*, *LhTRXh9*, *LhTRXh2*, *LhTRXh2.2*, *LhTRX-h3*, *LhTRXf*, *LhTRXm1*, *LhTRXy2*, *LhNrx1.4*, *LhTRXh4*, *LhTRXh1*, *LhTRXm3*, *LhTRXh5*, *LhTRXlike1.1*, with 41.2% belonging to the *LhTRXh* subclass and 58.8% to other subtypes of genes. The expression of seven genes reached peak values under drought stress at 1d, including *LhTDX*, *LhLillium1*, *LhNrx1.3*, *LhNrx1.6*, *LhNrx1.1*, *LhTRXlike1*. The peak time of expression varied among different *LhTRXs* under high-temperature stress, reaching peak values at 1 h, 3 h, 12 h, 1 d, and 3 d, respectively, with the proportion of gene numbers being 33.3%, 19%, 9.5%, 16.6%, and 9.5%, respectively. The expression levels of *LhTRXlike1*, *LhTRXlike1.1*, and *LhTRXlike2* genes gradually decreased with increasing stress time. It is possible that these three genes do not function under high-temperature stress. Under drought stress, 20 genes reached their peak values at 3 d of stress, including *LhTRXx*, *LhTRXm2*, *LhTRXm2.1*, *LhTRXm4*, *LhLillium1.1*, *LhTRXlike1*, *LhTRXh2.2*, *LhTRXh4*, *LhClot*, *LhTRz*, *LhNrx1.2*, *LhTRXm3*, *LhHCF164*, *LhTRXm1*, *LhTRXy2*, *LhTRXh2*, *LhTRXh3*, *LhTDX*, *LhTRXf*, *LhTRXy2*. And nine genes reached their peak values at 12 h of stress, including *LhTRXh5*, *LhNRX1.5*, *LhNRX1.7*, *LhNRX1.3*, *LhTRXh1.1*, *LhNRX1.4*, *LhNRX1.6*, *LhTRXlike2*, *LhLillium1*, among which the *LhTRXh* subclass accounted for 20.6%, indicating that the *LhTRXh* subclass played a certain role under abiotic stress.

### 2.6. Expression Patterns of Genes under Drought Stress

The gene expression profile revealed that *LhTRXs* exhibited a robust expression response under drought stress, and many genes were found to possess drought-related response elements in the *cis*-acting element analysis. Therefore, we further analyzed the gene expression patterns under drought stress by RT-qPCR and found two different expression patterns. Firstly, *LhTRXh2*, *LhTRXh3*, *LhNRX1.6*, *LhNRX1.7*, and *LhClot* genes showed high expression levels at 12h after the onset of drought stress, while only *LhTRXh2.2* genes exhibited high expression levels at 3 days after drought stress (Figure 6). This suggests that the stress response is induced by the interaction of different transcription factors, leading to varied temporal expression patterns among the *LhTRX* genes. The combination of transcriptome analysis and RT-qPCR results revealed a more pronounced upregulated expression of *TRXh*. Previous studies have shown that *TRX-h3* is significantly upregulated in stress-tolerant varieties [19], and therefore, *TRX-h3* was selected for the follow-up study.

### 2.7. Subcellular Localization

In order to determine the subcellular localization of *LhTRX-h3*, we cloned the complete coding region sequences of *LhTRX-h3* and inserted them into the pJIT166-GFP vector to overexpress *LhTRX-h3*-*GFP* fusion proteins under the control of a 35S promoter. The results demonstrated that fluorescence signals were detected in the cytoplasm, consistent with the predicted results (Figure 7).

### 2.8. LhTRX-h3 Reduces Callus Oxidative Stress

In order to explore the role of *LhTRX-h3* in regulating REDOX homeostasis under drought stress, we constructed overexpression and CRISPR vector (Appendix A). WT stands for wild-type callus that has not been genetically modified, *LhTRX-h3*-OE stands for overexpressed strain-positive callus, and *LhTRX-h3*-KO stands for positive callus of CRISPR vector transgene. Callus samples were exposed to 15% PEG6000 stress, and ImajeJ analysis was conducted to measure callus surface area (Figure 8A,B), The results revealed that WT and *LhTRX-h3*-KO callus grew slowly under stress, with the callus color becoming darker and granulated. Notably, the increase in callus surface average area of *LhTRX-h3*-OE-lines was 2.21 and 1.52 folds higher over WT and KO-lines, and the mean weight of per callus mass was 3.92 and 2.26 folds higher in the *LhTRX-h3*-OE compared to the WT and *LhTRX-h3*-KO (Figure 8C), respectively. Conversely, overexpressed positive callus exhibited normal callus appearance and fine growth under stress, with the fastest weight gain observed.

H_2_O_2_ and MDA content are crucial parameters to reflecting the potential antioxidant capacity of an organism [27]. Therefore, we tested a total of 0.1 g of randomly picked callus as one sample and used kits to detect H_2_O_2_ and MDA levels in the samples. The results indicated that the H_2_O_2_ (Figure 9B) and MDA (Figure 9C) contents in overexpressed callus were the lowest at day 0, followed by the wild type, while the *LhTRX-h3*-KO exhibited the highest contents. After 30 days of stress, both H_2_O_2_ and MDA contents increased rapidly, with MDA levels showing a more significant increase. MDA content remained lower in *LhTRX-h3*-OE compared to WT and *LhTRX-h3*-KO, indicating less damage in the *LhTRX-h3*-OE strain. DAB reacts rapidly with H_2_O_2_ in plant tissues to form a reddish-brown reagent that localizes H_2_O_2_ in the tissue. To ensure the reliability of the data, DAB staining was also performed, and the results were consistent with the previous findings. In conclusion (Figure 9), these results suggest that *LhTRX-h3* may alleviate growth restrictions during stress, regulate REDOX levels in vivo, and enhance tolerance to abiotic stress. In addition, we also quantified *LhTRX-h3* (Figure 9D), and the results showed that the expression level of *LhTRX-h3* increased significantly after stress, and the quantification of its homologous gene *LhTRXh2.2* (Figure 9E) found that the expression level of *LhTRX-h3*-KO line also increased significantly, so there may be gene redundancy.

## 3. Discussion

TRX proteins play a pivotal role in defense against oxidative stress and serve as key regulatory elements in defense mechanisms [28]. ROS generation and scavenging are closely related to the regulation of plant immune signaling pathways [29,30]. Through genome-wide analysis of thioredoxin in *L. chinense*, 42 *LcTRX* genes were identified, which was consistent with the number found in bread wheat (*Triticum aestivum* L.; *Ta*) but differed from that of Arabidopsis (41), Foxtail millet (*Setaria italica*) (35), and poplar (49) [31,32,33,34,35]. These *LcTRX* genes were categorized into 15 subfamilies: *TRXh*, *TRXf*, *TRXm*, *TRXz*, *TRXx*, *TRXy*, *TRXo*, *TRXlike1*, *TRXlike2*, *TRXNrx*, *TRXPicot*, *TRXclot*, *CDSP32*, *HCF164*, and *TRXLillium*. Among these, the h subclass comprised 10 TRX protein sequences, further subdivided into three classes, making it the largest subclass of TRX. This distribution characteristic of TRX in plants aligns with the results observed in species [36].

Analysis of conserved gene domains revealed that TRX proteins with the same domains belonged to the same subclass, suggesting similar functions. In the case of *LcNRX*, the presence of multiple Motif1, representing the conserved domains of WCGPC, may be attributed to its larger protein size and specialization [17]. However, other atypical TRX proteins, despite being larger, contain only one Motif1, implying that other conserved domains may have been lost during evolution. Gene duplication is a significant factor in organismal evolution, facilitating the generation of new adaptive functions, contributing to the emergence of new traits, and expanding gene families [37]. This duplication process includes segmental duplication and tandem duplication [38]. In the *LcTRX* gene family, we identified three fragment duplication events. Ka/Ks analysis of these three pairs of fragment duplication events revealed that all values were less than 1, indicating that evolution was subjected to purification selection.

Thioredoxin typically transfers reducing equivalents to oxidized proteins through dimercapto-disulfide bond exchange reactions [39]. Increasing evidence underscores the importance of thioredoxin (TRX) in the REDOX control of plant metabolism [15].

Overexpression of *TRX* has been shown to enhance plant stress resistance under abiotic stress. Studies have demonstrated that overexpression of *MaCDSP32* in mulberry confers drought resistance in transgenic plants [40]. Similarly, overexpression of *TRX-h2* in Arabidopsis enhances plant tolerance to cold environments [41]. However, in the context of water scarcity, it was found that *TRXy* mutant and double mutant plants exhibited lower stress tolerance [42]. Therefore, it is speculated that *LcTRXs* have the potential to improve stress resistance under abiotic stress conditions. We conducted an analysis of the expression patterns of *LhTRXs* under cold, heat, and drought stress to gain insight into the cross-regulatory functions of *LhTRXs* under various stress conditions. Our findings revealed varied expression patterns among *LhTRX* genes, with some genes showing high expression levels under all three stresses, while others were predominantly expressed under one or two stress conditions. For instance, *LhTRX-h2.2* was exclusively expressed under cold and drought stress, while *LhTDX* exhibited higher expression levels only under cold stress. On the other hand, *LhTRXf*, *LhTRXm2*, *LhTRXm4*, and *LhTRX-h3* were all expressed under the three stress conditions of cold, heat, and drought. Notably, the expression level of *LhTRX-h3* was particularly high under stress conditions. Therefore, in abiotic stress, *LhTRX-h3* may co-regulate abiotic stress with other genes. Additionally, qRT-PCR analysis conducted at 0 h, 1 h, 12 h, and 3 days under drought stress revealed two distinct expression patterns, namely high expression at 12 h and high expression at 3 d, respectively. It is speculated that these different expression patterns may be associated with oxidative stress, wherein temporal differences in the binding of various transcription factors to downstream genes contribute to the stress response. Drought stress often triggers the excessive accumulation of ROS, particularly O_2_^−^ and H_2_O_2_, leading to significant oxidative damage in plants [43,44]. To investigate the effects of thioredoxin (TRX) overexpression and knockout on ROS levels, we constructed overexpression and knockout vectors. Our results indicated that the growth rate of *LhTRX-h3*-overexpressing callus under stress was faster compared to WT and knockout, and the in vivo ROS content was also lower [45], consistent with previous findings [44,46]. In summary, this study identified a drought-responsive gene, *LhTRX-h3*, through genome-wide identification and bioinformatics analysis of the *L. chinense TRX* gene family, combined with expression pattern analysis. Further overexpression and knockout of *LhTRX-h3* in *L.* hybrid, combined with drought stress treatment and physiological and biochemical indexes, revealed that the overexpression of *LhTRX-h3* was able to reduce the accumulation of ROS in *L.* hybrid under drought stress, and this study reveals the function of *LhTRX-h3* in drought tolerance in *L.* hybrid. The overexpression of *LhTRX-h3* was found to reduce the accumulation of reactive oxygen species under drought stress, providing a candidate gene for the improvement of drought tolerance in *L.* hybrid, which is of great significance for the advancement of molecular breeding of *L.* hybrid for resistance to drought, and we will investigate the function of *LhTRX-h3* in seedlings and the molecular regulatory mechanism under drought stress in depth.

## 4. Materials and Methods

### 4.1. Identification of TRX Gene in L. chinense

To identify the *TRX* gene in *L. chinense*, 39 typical and atypical TRX protein sequences of *Arabidopsis thaliana* were downloaded from the TAIR database (https://www.arabidopsis.org/Blast/index.jsp) (accessed on 30 October 2021). From the pfam website (http://pfam-legacy.xfam.org/) (accessed on 3 November 2021), the TRX hidden Markov number is PF00085, and this number has been used as the query condition. Blastp and HMMER were used to query the target sequence in the *L. chinense* protein database, and 70 candidate sequences were obtained. Based on the TRX conserved domains, CDD-search (https://www.ncbi.nlm.nih.gov/Structure/cdd/wrpsb.cgi) (accessed on 3 November 2021) was used to check the conserved domains of candidate sequences to further screen out redundant sequences, and finally obtain the target sequence. Gene properties, including length, molecular weight, and isoelectric point of each protein, were determined using the ExPASy website (https://web.expasy.org/protparam) (accessed on 3 November 2021) tool. Subcellular localization of *LcTRX* genes was predicted by Cell-PLoc 2.0 (www.csbio.sjtu.edu.cn/bioinf/Cell-PLoc-2) (accessed on 5 November 2021).

### 4.2. Phylogenetic Analysis and Conserved Domains and Gene Structure Analysis

Rice, grape, Arabidopsis, and poplar TRX protein sequences used to construct the phylogenetic tree were downloaded from Phytozome. Multiple sequence alignment of *TRX* gene family members was performed using MAFFT software (https://mafft.cbrc.jp/alignment/software) (accessed on 15 November 2021) with default parameters. MAGE 7.0 was utilized to construct the phylogenetic tree, employing the neighbor-joining method with a bootstrap value of 1000 to analyze the evolution of the *TRX* gene in *L. chinense*. The phylogenetic trees were beautified using Evolview online software (V3.0) (http://www.evolgenius.info/evolview) (accessed on 25 November 2021). The conserved motifs of LcTRX proteins were analyzed by the online software MEME (V5.5.0) [47]. The exon and intron structure features of each LcTRX protein gene were obtained from the genome file (https://www.ncbi.nlm.nih.gov/assembly/GCA_003013855.2#/st) (accessed on 4 December 2021), and the motifs and gene structure were visualized using TBtools software (V2.096) (https://github.com/CJ-Chen/TBtools/releases) (accessed on 10 December 2021).

### 4.3. Chromosomal Location and Collinearity Analysis

The chromosomal location of the *LcTRX* family was determined using the *L. chinense* genome-wide database. Gene duplication events and collinearity between *L. chinense*, Arabidopsis, and rice were visualized using TBtools. Ka/Ks for each gene pair were calculated using KaKs Calculator 2.0 [48].

### 4.4. Cis-Acting Element Analysis

The upstream 2000 bp promoter sequence of the *LcTRX* gene was extracted from the whole genome sequence of *L. chinense*. Predictions for *cis*-acting elements were obtained using the online site PlantCARE (http://bioinformatics.psb.ugent.be/webtools/plantcare/html/) (accessed on 13 December 2021).

### 4.5. Plant Material and qRT-PCR

To study the response of the *TRX* gene family under drought stress, three replicates of leaves from *L.* hybrid seedlings treated with drought stress for 0 h, 1 h, 12 h, and 3 days were collected. For the drought stress treatment, PEG6000 with a concentration of 15% was poured into the soil of potted seedlings once every two days. Total RNA was isolated from the leaves using the FastPure Total RNA Isolation Kit (Vazyme, Nanjing, China). The quality of the extracted RNA was validated using agarose gel electrophoresis and a Nanodrop ND-1000 spectrophotometer. Subsequently, a HiScript^®^III 1st Strand cDNA Synthesis Kit (Vazyme, Nanjing, China) was used to synthesize cDNA with the extracted RNA as a template. According to the results of RNA-seq, 6 genes with obvious expression trend were selected for transcriptomic verification under drought stress, and primers were designed using Oligo 7 (v:7.56). Real-time fluorescence quantification was performed using a Roche LightCycler 480II instrument with AceQ qPCR SYBR Green Master Mix (Without ROX) reagents. The results were calculated using the 2^−ΔΔCT^ method [26,49].

To investigate the role of *LhTRX-h3* under drought stress and to probe the redundant role of *TRXh2.2* in knockout vectors, we used transgene-positive callus samples exposed to drought stress for 0 and 30 days to extract RNA, reverse transcribe it, and perform real-time fluorescence quantification using the above-mentioned methodology and instrumentation. At least three biological replicates were performed for each time point.

### 4.6. Subcellular Localization

To verify the subcellular localization of *TRX-h3*, we obtained the *LhTRX-h3* target fragment by PCR using the *LcTRX-h3* sequence as the reference sequence and the cDNA of *L.* hybrid plants as the template. The plasmid pJIT166-GFP was digested with XbaI and BamHI enzymes, and the linear vector fragment was ligated with the target gene fragment to construct the p35S:TRX-h3-GFP fusion expression vector. Plasmids were extracted using an endotoxin-free plasmid extraction kit (DP117, TIANGEN). The p35S:H2B-mCherry, serving as a control for nuclear localization signal, and the recombinant vector were co-transfected into protoplasts via PEG-mediated transient transformation. After 3 days, GFP expression was observed using a laser confocal microscope (LSM 800 system; Zeiss, Germany).

### 4.7. LhTRX-h3 Cloning and Vector Construction

The full-length coding sequences (CDSs) of *LhTRX-h3* were amplified by PCR and then cloned into the PBI121 vector digested with ScaI and XbalI under the control of the CaMV 35S promoter for overexpression of *LhTRX-h3*, respectively [50]; the following is abbreviated as *LhTRX-h3*-OE. After constructing the vector, *LhTRX-h3*-OE was transformed into agrobacterium-competent cells EHA105, and single clones were picked to verify their correctness before infection. The positive callus was screened by genomycin and verified by PCR and sequencing. To generate *LhTRX-h3*-KO, CRISPR-P v2.0 (CRISPR-P v2.0 (hzau.edu.cn) was utilized to identify sequences that matched the target, and primer design was performed using primer design (http://skl.scau.edu.cn/primerdesign/) (accessed on 13 October 2022). CRISPR vector was constructed using Liu’s method [51]. The vector was transformed into the callus by *Agrobacterium* mediation [52]. Positive callus was obtained through 70 mg/L genomycin screening, and mutants identified via PCR and sequencing were used for subsequent experiments.

### 4.8. p35S:LhTRX-h3 and Positive Callus Stress Treatment

Based on the expression profile under cold, heat, and drought stress, we chose to subject *LhTRX-h3*-OE- and *LhTRX-h3*-KO-positive callus to drought stress treatment. Three strains of positive callus and vector were chosen, with each strain consisting of 5 dishes, and each dish containing eight callus samples. Additionally, there were three dishes for the wild type. For the drought stress treatment, the basal growth medium prepared with 15% PEG6000 was utilized. Both treatments were administered for 30 days, and pictures were taken at 0 and 30 days to measure the increase in callus surface area and weight.

### 4.9. DAB Staining and Physiological Index Detection of LhTRX-h3-OE- and LhTRX-h3-KO-Positive Callus under Drought Stress

Callus treated with 15% PEG6000 for 0 and 30 days was incubated with 1.0 mg/mL 3-diaminobenidine (DAB, dissolved in distilled water, pH 3.8) for 12 h [14,53]. In addition, H_2_O_2_ (Beyotime, Shanghai, China) and MDA (Jiancheng, Nanjing, China) levels were determined after stress treatment using the kit method as described in the kit instructions, with at least five replicates for each sample.

## Figures and Tables

**Figure 1 plants-13-01674-f001:**
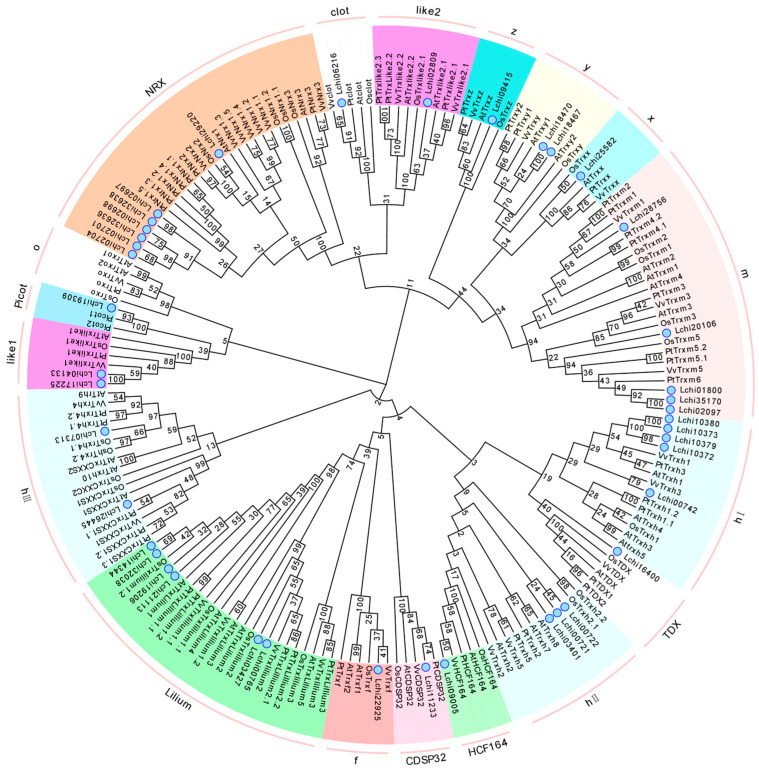
Phylogenetic trees of *L. chinense* and Arabidopsis, rice, grape, and poplar. Different colors represent different subclasses, and the blue circles represent *L. chinense*. Numbers on branches indicate percent reliability of bootstrap values based on 1000 replicates.

**Figure 2 plants-13-01674-f002:**
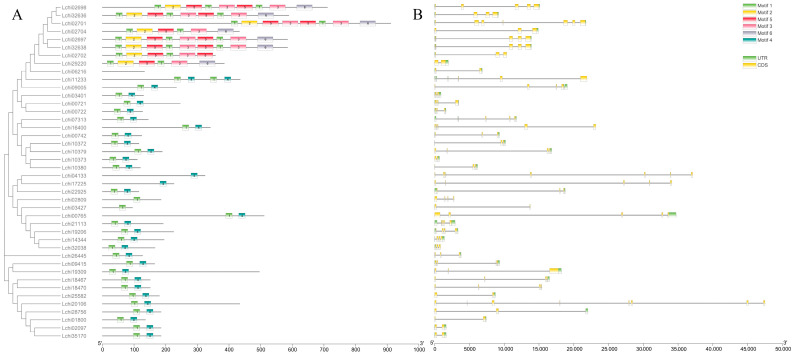
Analysis of conserved motif elements and gene structures of *LcTRXs*. (**A**) Distribution of conserved motif elements. (**B**) Gene structures; scale markers represent gene length (bp) and protein sequence length (aa).

**Figure 3 plants-13-01674-f003:**
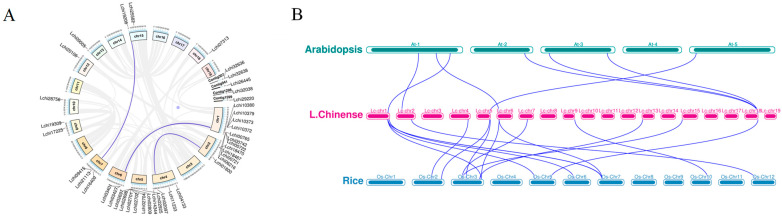
Genome-wide synteny analysis of *TRX* gene family among *L. chinense* and other two species. (**A**) Distribution and fragment duplication of the *LcTRX* gene in *L. chinense*. The circles are chromosomal genes, and the names are displayed outside the circles. (**B**) Species collinearity analysis of *L. chinense*, Arabidopsis, and rice. Grey lines in the background indicate collinear regions in *L. chinense* and other plant genomes, while blue lines highlight shared *TRX* genes.

**Figure 4 plants-13-01674-f004:**
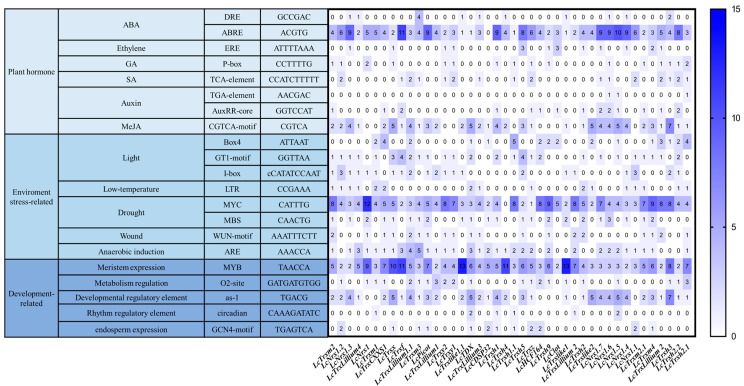
Results of predicted elements from the PlantCare database for the LcTRXs gene promoter. The vertical axis shows the element types, categorized into plant hormones (light blue), environmental stress-related elements (blue), and growth and development-related elements (dark blue). Blue boxes on the right indicate that the element has a high predicted presence in the LcTRXs gene promoter, while white boxes indicate that the element is absent or present in low amounts in the gene promoter.

**Figure 5 plants-13-01674-f005:**
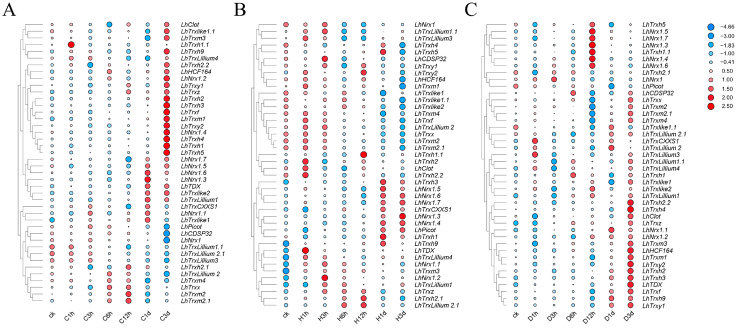
Transcriptional expression patterns of *TRX* genes in *L.* hybrid under (**A**) cold, (**B**) heat, and (**C**) drought stress are depicted. The *LhTRXs* were subjected to three different stress factors: cold, heat, and drought stress. The designations Heat_0h, Heat_1h, Heat_3h, Heat_6h, Heat_12h, Heat_1d, and Heat_3d represent three biological replicates for each time point (0 h, 1 h, 3 h, 6 h, 12 h, 1 day, 3 days). Transcript abundance levels are represented using the log2(FPKM + 1) transformation. The values on the right panel of the heatmap indicate the expression level.

**Figure 6 plants-13-01674-f006:**
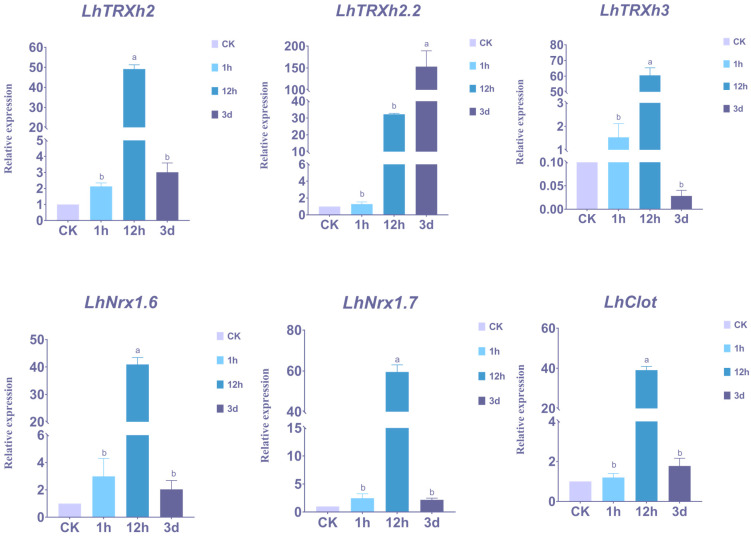
The expression of 6 *TRX* genes in drought-treated samples was analyzed by qRT-PCR. Vertical bars represent standard deviation, with 18S rRNA gene serving as the internal reference gene, and each sample was repeated three times. The data were subjected to analysis using a one-way ANOVA and multiple comparisons employing the Duncan method (*p* < 0.05). Different letters represent significant differences (a, b).

**Figure 7 plants-13-01674-f007:**
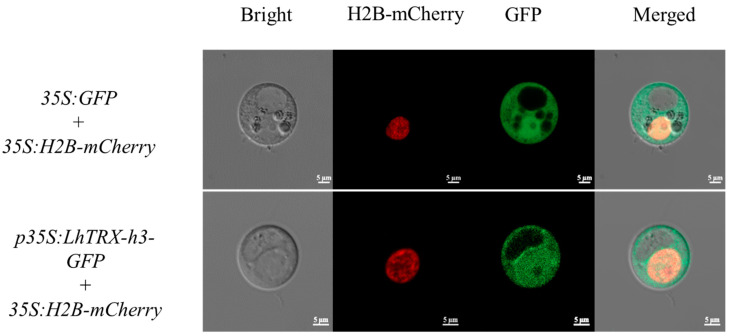
Subcellular localization of LhTRX-h3. The first panel indicates bright field, H2B-mCherry (second panel) indicates red fluorescence photography GFP indicates (third panel) green fluorescence photography, and the merged fourth panel indicates fusion of red fluorescence, green fluorescence, and bright field. Bar = 5 μm.

**Figure 8 plants-13-01674-f008:**
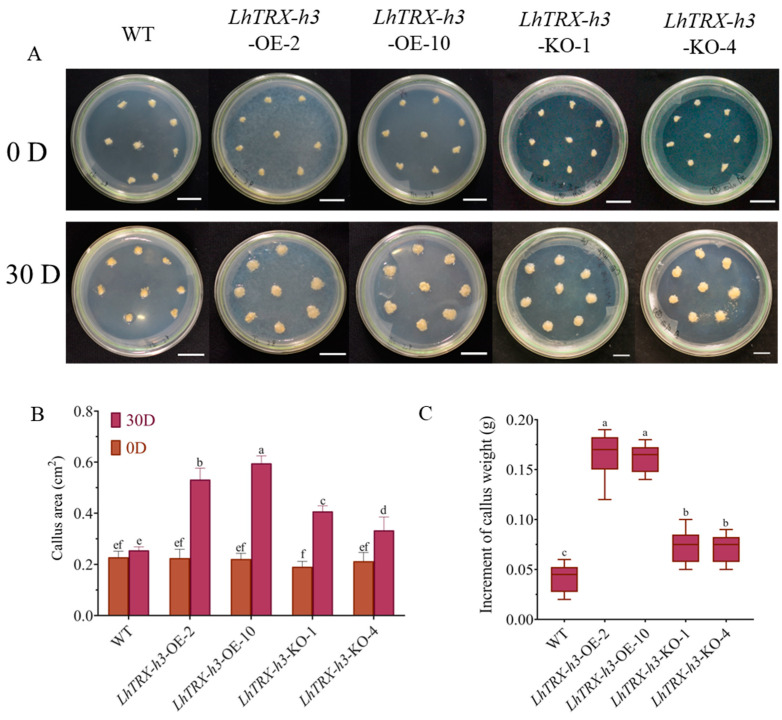
Analysis of healing growth traits under drought stress. (**A**) Callus of 0 and 30 d treated with 15%PEG6000. (**B**) Statistics of callus area at 0 and 30 days under 15%PEG6000 stress (*n* = 40, *p* < 0.05). (**C**) The difference between stress 0 d and 30 d callus weight (*n* = 40, *p* < 0.05). The data were subjected to analysis using a one-way ANOVA and multiple comparisons employing the Duncan method; different letters mean that they differ in significance (a–f).

**Figure 9 plants-13-01674-f009:**
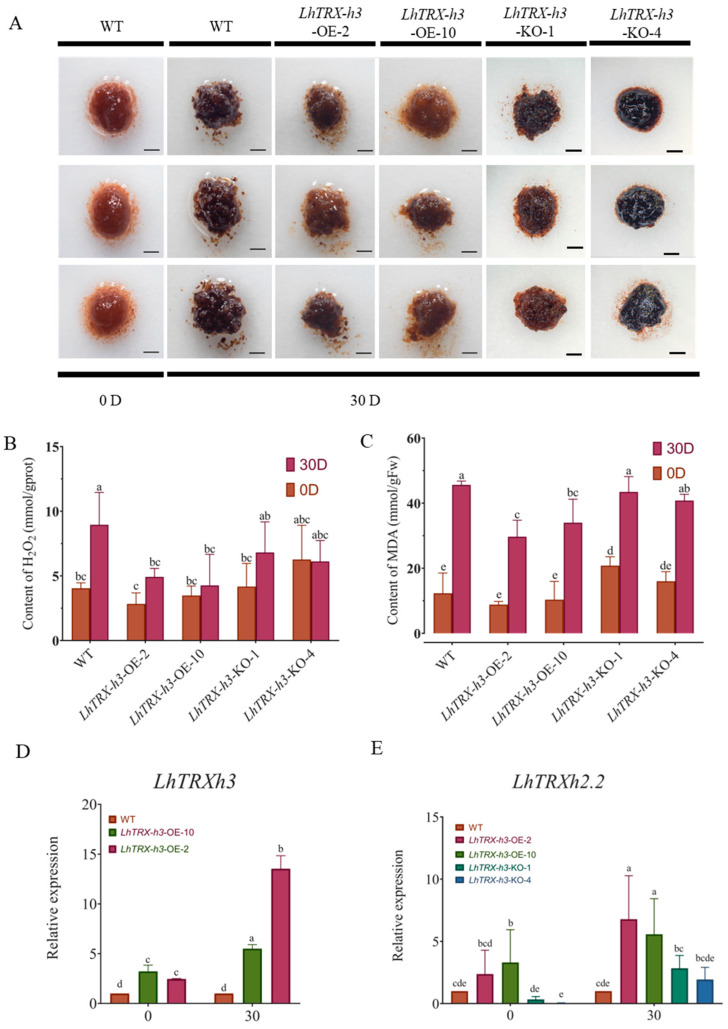
Physiological index determination and gene quantitative verification. (**A**) Callus treated with 15%PEG6000 at 0 and 30 d were stained with DAB. Bar = 2 mm. (**B**) Callus treated with 15%PEG6000 on 0 and 30 d were determined by H_2_O_2_. (**C**) The callus treated with 15%PEG6000 at 0 and 30 d were determined by MDA (*n* = 5, *p* < 0.05). (**D**) Expression of *LhTRX-h3* under drought stress. (**E**) Expression of *LhTRXh2.2* under drought stress (*n* = 3, *p* < 0.05). Statistically significant differences were determined using one-way ANOVA and multiple comparisons employing the Duncan method; different letters mean that they differ in significance (a–e).

**Table 1 plants-13-01674-t001:** Summary of *LcTRXs* gene family characters.

TRX Class	TRX Subclass	Gene ID	Gene Name	aa	Mw (kDa)	pI	Putative Localization
Typical TRX	HI	*Lchi10380*	*LcTRXh1*	119	13.523	8.95	Cytoplasm/Microsome/Mitochondrion
		*Lchi10373*	*LcTRXh1.1*	109	12.247	7.67	Vacuole
		*Lchi00742*	*LcTRXh3*	123	13.601	6.05	Cytoplasm
		*Lchi10379*	*LcTRXh4*	188	21.001	5.31	Cytoplasm
		*Lchi10372*	*LcTRXh5*	114	12.688	5.06	membrance
	HII	*Lchi03401*	*LcTRXh2*	129	14.665	5.40	Secreted protein
		*Lchi00721*	*LcTRXh2.1*	245	26.973	8.33	Cytoplasm/Nucleus
		*Lchi00722*	*LcTRXh2.2*	126	14.386	6.29	membrance
	HIII	*Lchi07313*	*LcTRXh9*	144	15.827	4.70	Cytoplasm
		*Lchi26445*	*LcTRXCXXS1*	126	14.376	5.14	Nucleus
	f	*Lchi22925*	*LcTRXf*	114	12.701	8.74	Cell wall; Cytoplasm
	m	*Lchi01800*	*LcTRXm4*	135	15.104	4.89	Nucleus
		*Lchi35170*	*LcTRXm2*	184	19.997	8.92	Mitochondrion
		*Lchi02097*	*LcTRXm2.1*	184	20.025	9.05	membrance
		*Lchi20106*	*LcTRXm3*	433	47.513	7.54	Cytoplasm/Mitochondrion
		*Lchi28756*	*LcTRXm1*	184	19.907	9.47	membrance
	x	*Lchi25582*	*LcTRXx*	179	19.881	9.07	Secreted protein
	y	*Lchi18467*	*LcTRXy1*	150	17.096	4.52	Cytoplasm
		*Lchi18470*	*LcTRXy2*	150	17.123	4.64	Microsome
	z	*Lchi09415*	*LcTRXz*	164	18.553	5.59	Nucleus
Atypical TRX	like1	*Lchi04133*	*LcTRXlike1*	323	36.291	5.72	Cytoplasm
		*Lchi17225*	*LcTRXlike1.1*	225	25.826	4.70	Endoplasmic reticulum
	like2	*Lchi02809*	*LcTRXlike2*	184	21.079	7.68	Endoplasmic reticulum
	Lillium	*Lchi03427*	*LcTRXLillium 2*	94	10.539	5.20	Cytoplasm
		*Lchi00765*	*LcTRXLillium 2.1*	510	56.397	5.00	membrance
		*Lchi19206*	*LcTRXLillium1*	224	24.878	9.30	Nucleus
		*Lchi21113*	*LcTRXLillium1.1*	191	20.942	8.45	Nucleus
		*Lchi14344*	*LcTRXLillium3*	194	21.648	8.42	membrance
		*Lchi32038*	*LcTRXLillium4*	165	18.372	6.94	Chloroplast
	CDSP32	*Lchi11233*	*LcCDSP32*	434	49.556	8.45	Cytoplasm
	Clot	*Lchi06216*	*LcClot*	132	14.919	6.90	Chloroplast/Cytoplasm
	HCF164	*Lchi09005*	*LcHCF164*	233	25.733	5.10	Cytoplasm
	Nrx	*Lchi29220*	*LcNrx1*	384	43.484	5.19	Centriole
		*Lchi02697*	*LcNrx1.1*	584	65.705	4.74	Mitochondrion
		*Lchi32638*	*LcNrx1.2*	584	65.705	4.74	Chloroplast
		*Lchi32636*	*LcNrx1.3*	588	66.549	4.98	Chloroplast
		*Lchi02698*	*LcNrx1.4*	710	79.859	5.00	Mitochondrion
		*Lchi02701*	*LcNrx1.5*	910	102.664	5.24	Mitochondrion
		*Lchi02702*	*LcNrx1.6*	356	40.369	4.96	Mitochondrion
		*Lchi02704*	*LcNrx1.7*	432	49.007	5.79	Mitochondrion
	picot	*Lchi19309*	*LcPicot*	495	54.049	5.07	Mitochondrion
	TDX	*Lchi16400*	*LcTDX*	340	38.398	4.85	Nucleus

## Data Availability

The data and results are available to every reader upon reasonable request.

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
