# Peer review of "Unraveling the Role of the Liriodendron Thioredoxin (TRX) Gene Family in an Abiotic Stress Response"

_plants, 2024, doi:10.3390/plants13121674_

Round 1
Reviewer 1 Report
Comments and Suggestions for Authors
The manuscript “
Unraveling the Role of Liriodendron Thioredoxin (TRX) Gene Family in Abiotic Stress Response” by Lu et al. describes the computational and experimental characterization of the TRX gene family in Lirodendron in response to drought, cold, and heat stress. I would have liked to see stronger arguments why Lirodendron is important.
Authors did a range of experiments, including qRT-PCR, cellular localization, and overexpression and knockouts of TRX genes in calluses. Testing overexpression and knockout calluses for H2O2 and MDA content in response to stress was certainly interesting, though both the Results and Methods sections would benefit from adding more information on how the various experiments were performed (I address specific points below). The English language is fine, except for minor revisions.
Specific points
INTRODUCTION
“While the TRX family has been extensively studied in Arabidopsis, rice, and wheat, its exploration in Liriodendron remains unreported, leaving its biological function largely unknown.” Rephrase. Please consider that if the TRX family has been extensively studied in other plants, one can infer that they have similar biological functions in Liriodendron.
“based on the Liriodendron hybrids transcriptome data and RT-qPCR experiments” add “publicly available” to make clear that the data was not generated in this study
“balance of protein dimercaptan/disulfide bonds balance” remove one “balance”
“and TRXh overexpressing plants in tomato” how about “TRXh overexpressing tomato plants”
“had lower MDA levels” explain MDA at first mentioning
“it is important to explore the abiotic stress tolerance of TRX” rephrase, e.g. “…the role of TRX in abiotic stress tolerance.”
“and quantitatively explored the expression pattern of TRXs under cold, heat, and drought stress by real-time fluorescent light.” Did you mean by RT-qPCR?
“which provided a more theoretical basis for” why “more theoretical”? how about “which provided a basis for the further study of TRXs in Liriodendron. “
RESULTS
“unrooted phylogenetic tree …” mention the tree-generating method (neighbor-joining; it is in the Methods section, but as reader, I want to know this when I look at the tree)
Figure 1.: “L.Chinense” should be L. chinense”
“Arabidopsis” should not be italicized (throughout the paper)
“Here, I calculated the Ka/Ks” use “we” rather than “I”
“Arabidopsis and rice belong to monocotyledonous and dicotyledonous plants, respectively.” Order is switched
“This suggests that L. chinense is more closely related to dicotyledonous plants like Arabidopsis than to monocotyledonous plants like rice in the evolutionary process” give the reader a little context of what is known about this species evolutionary relationship; is it generally thought to be closer to eudicots? Citation?
“Moreover, than half of the genes” Did you mean “More than half…”?
“The results reveal that the number of cis-acting elements of TRXs was large and the types were rich.” “types were rich”? rephrase
Figure 4. it would be helpful to include the motifs, so the reader can gauge the likelihood of random occurrence
“In order to study the expression of TRXs under cold, heat, and drought stresses in L. hybrids, the laboratory's previous RNA-seq data” add citation
Figure 5.
“Transcript abundance levels are represented using the log2(FPKM + 1) transformation. The values on the right panel of the heatmap indicate the expression level.”
Based on the negative values, it looks like the colors on the right indicate fold change (log2FC) rather than log2(FPKM +1).
“Therefore, we further analyzed the gene expression pattern under drought stress” add what method you used (qRT-PCR).
“exhibited highly expression levels at 3 days” change highly to high
Figure 6.
“with 18s serving…” change to “with the 18S rRNA gene serving…”
Figure 8.
Why stacking bars on top of each other? It seems to skew the differences if a callus starts smaller and grows less. Is there any good way to normalize for the difference in callus size? Or at least show the bars next to each other, rather than stacked on top of each other.
“Therefore, we detected the level of H2O2 and MDA in callus samples” Mention briefly how you did this.
“However, it was observed that the increase in MDA content was less pronounced in the LhTRX-h3-OE strain compared to the WT and LhTRX-h3-KO, suggesting that their cells were not more severely damaged.” Too many double-negatives makes this difficult to follow, clarify. Perhaps something like “MDA content remained lower in LhTRX-h3-OE compared to WT and KO, indicating less damage in the OE strain.”
“Although there was no significant increase in H2O2, LhTRX-h3-OE 242 demonstrated a superior ability to reduce ROS content in vivo.” Explain, how did you test this?
“To ensure the reliability of the data, DAB staining was also performed,” Remind the reader of the purpose of DAB staining
“In addition, we also quantified LhTRXh3 (Fig. 9D)” add what you quantified (expression)
Figure 9
“(B) …were determined by hydrogen peroxide,” do you mean you determined relative content of hydrogen peroxide”? clearly state what you determined, and briefly mention with what method.
Figure 9 B, C, again, avoid stacking bars; it does not seem objective.
DISCUSSION
“42 LcTRX genes were identified, which was consistent with the number found in Bread wheat but differed from that of Arabidopsis, grape, and poplar[33-36]” mention the numbers within those species
“it was found that TRX-y monotonic and bitid plants” explain (in the text, not to me)
“Therefore, it is suggested that in abiotic stress, LhTRX-h3 may serve as a major regulatory gene” why does response to abiotic stress suggest a “regulatory” role? It could be just part of a response, not necessarily a regulator.
“Moreover, the quantitative analysis of LhTRXh3 gene found that LhTRX-h3-OE strains significantly increased after stress, indicating that LhTRXh3 may play its function under stress and lead to a significant increase in its expression level.” Circular, increased expression indicates increased expression. Also, this already mention in a previous paragraph.
I suggest you end the discussion with some summarizing statement.
MATERIALS & METHODS
“Real-time fluorescence quantification was performed using previously described instruments and reagents.” Add citation for “previously described”
“The homologous genes of LhTRXh3 were quantified to confirm gene redundancy in the CRISPR vector” clarify
“The vector was transferred into the callus by Agrobacterium mediated” add “transformation
“Positive callus was obtained throught genomycin screening” mention genomycin concentration
“Additionally, the content of H2O2 (Beyotime, China) and MDA (Nanjing Jiancheng, China) was measured after stress treatment…” How was the content of H2O2 and MDA measured?
Comments on the Quality of English Language
The English language is fine, except for minor revisions.
Author Response
We gratefully thank the editor and all reviewers for their time append making their constructive remarks and useful suggestions, which has significantly raised the quality of the manuscript and has enable us to improve the manuscript. Each suggested revision and comment, brought forward by the reviewers was accurately incorporated and considered. Below the comments of the reviewers are response point by point and the revisions are indicated.

Reviewer 2 Report
Comments and Suggestions for Authors
The manuscript (MP) “Unraveling the Role of Liriodendron Thioredoxin (TRX) Gene Family in Abiotic Stress Response” submitted by Tong et al. investigated the whole TRX Gene Family in Liriodendron, and analyzed the subcellular localization and gene function of LhTRXh3. Overall, this manuscript appears suitable for publication.
Small flaws:
|
The author did not elaborate well on why only LhTRXh3 was analyzed instead of other LcTRXs, such as: LhTRXh2.2, LhTRXh2, LhNRX1.6 and so on. I suggest authors add some sentences in section 2.6 to explain the reason for selecting LhTRXh3 for further experiments. |
Author Response

(The authors gave the same response as above.)

Reviewer 3 Report
Comments and Suggestions for Authors
The article “Unraveling the Role of Liriodendron Thioredoxin (TRX) Gene Family in Abiotic Stress Response” written by Tong et al., is well-structured. The results are well-organized. However, I suggest elaborating on the discussion section a bit more. In this section, the authors might consider the following points: implications of the obtained results and unanswered questions along with the potential new hypotheses that arise from the study. Overall, I recommend minor revisions.
Point 1: Please change “I” to “we” in the following line: “Here, I calculated the Ka/Ks of three pairs of tandem repeats……..”
Point 2: Please modify Figure 3.
Point 3: Please modify the figure 4. If possible, avoid green and red.
Point 4: Figure 5 is also of poor quality.
Comments on the Quality of English LanguageMinor editing of the English language would be helpful.
Author Response

(The authors gave the same response as above.)
